# NEEDLECHAIN:
# MEASURING INTACT LONG-CONTEXT REASONING CAPABILITY OF LARGE LANGUAGE MODELS

## ABSTRACT

The Needle-in-a-Haystack (NIAH) benchmark is commonly utilized to evaluate the capacity of large language models (LLMs) to manage long contexts by determining whether a model can identify query-relevant information amidst a vast amount of irrelevant text. This paradigm is increasingly regarded as a standard method for quantifying the effective context length of LLMs. However, we find that the context length measured in this manner does not accurately reflect the genuine context understanding capabilities of LLMs. Specifically, even advanced models like GPT-4o face challenges when the context includes only a few query-relevant sentences without irrelevant distractors. To address this, we introduce **NeedleChain**, a new benchmark designed to evaluate the range of context lengths that allow intact understanding by LLMs. Unlike NIAH, **NeedleChain** requires models to integrate and reason over the entire input to arrive at the correct answer. This benchmark is adaptable, enabling researchers to adjust both context length and reasoning order for a more thorough analysis of long-context performance. Experiments with various state-of-the-art LLMs reveal a notable gap between their ability to process long inputs and their capacity for full understanding, underscoring the need for benchmarks and methodologies beyond the NIAH paradigm. Additionally, we propose a straightforward yet effective strategy, ROPE Contraction, which directly enhances long-context reasoning without altering the architecture. Throughout this paper, we argue that instead of rapidly extending context length, improving comprehension within a limited range could be more advantageous.

## 1 INTRODUCTION

Over the past few years, we have witnessed significantly advancement of large language models (LLMs) in various aspects Yang et al. (2025a); Hurst et al. (2024); Team et al. (2025). Enhanced reasoning abilities now allow LLMs to tackle several challenges and expand the range of tasks they can address Liu et al. (2024a); Team (2025). In particular, there also has been significant progress in extending the maximum context length that LLMs can process to handle complex task Tworkowski et al. (2023); Ge et al. (2025); Jin et al. (2024); Ding et al. (2024). For instance, Llama-2 Touvron et al. (2023) accommodated a context size of 4,096 tokens, while two years later, Llama-4 Meta (2025) can manage up to 10 million tokens.

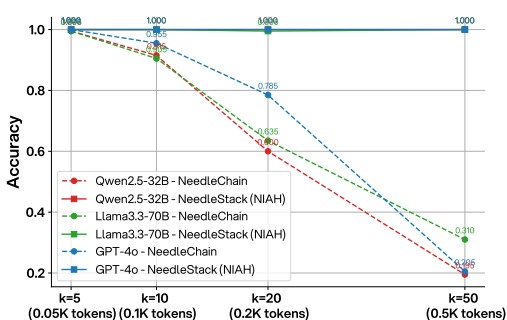

Figure 1: Performance comparison between the NeedleChain (Backward chain) and its parallel NIAH paradigm benchmark (NeedleStack). Reported number of tokens were estimated with Qwen2.5 tokenizer.

With extended context size of LLMs, the Needle-in-a-haystack (NIAH) benchmark has become a prominent tool for evaluating such long-context (LC) understanding capability of LLMs Laban et al. (2024); Schuster et al. (2025); Bianchi et al. (2025); Wang et al. (2024). It assesses how effectively

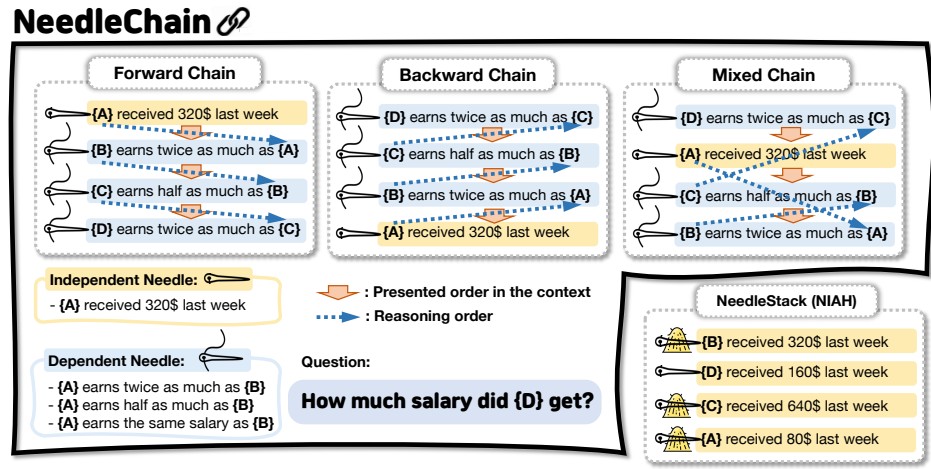

Figure 2: Performance variation with respect to the domain composition of training data

an LLM can locate a specific piece of query-relevant information (*"needle"*) embedded within a much larger body of text (*"haystack"*) which often contains irrelevant or distracting content. Within the NIAH paradigm, various studies have sought to measure the practical context size that LLMs can manage effectively An et al. (2024b); Kuratov et al. (2024). For example, Hsieh et al. (2024) reported that although Llama3.1 Grattafiori et al. (2024) is claimed to support a 128K length context, its effective context length is actually 32K.

However, we argue that NIAH paradigm based evaluation still tend to overestimate the actual context understanding capabilities of LLMs. NIAH includes substantial query-irrelevant information, which means that the LC understanding skills measured in this task are fundamentally different from those required for tasks that necessitates a thorough understanding of the entire context. Consequently, while NIAH serves as a useful baseline, it fails to provide a rigorous benchmark that effectively challenges and differentiates the advanced context comprehension abilities of state-of-the-art models An et al. (2024b).

To address this gap, we introduce NeedleChain, a new benchmark designed to rigorously assess holistic context comprehension. In NeedleChain, every piece of information is part of a causal chain and is indispensable for arriving at the correct answer; missing a single element makes a correct solution unattainable. We construct our benchmark from concise, interconnected statements, such as *"A received \$1,600 last week"* and *"A earns twice as much as B,"* which must be synthesized in concert to solve the given query.

In particular, we define a "**reasoning order**" concept and incorporate it into our benchmark. This refers to the logical order required to understand information within a context. We propose three variants: the forward chain (forces left-to-right comprehension), backward chain (forces right-to-left comprehension), and mixed chain. In addition to these three variants, we also create a NIAH dataset, dubbed **NeedleStack**, using the same benchmark component. Figure 2 provides specific examples and subsequent sections detail the construction process. By analyzing these variants, we identify previously overlooked vulnerabilities in LLMs' intact comprehension of the given context and report on their intact understanding capabilities.

Our findings reveal a significant deficiency in current models. Figure 1 contrasts the performance on NeedleChain (backward) benchmark with the NeedleStack (NIAH) benchmark. We reveal that the context understanding capability of LLMs dramatically deteriorates when the context consists solely of query-relevant, interconnected information. Notably, even with a context of just 200 tokens, models exhibit a significant failure to fully capture the required information. This directly casts a doubts on the reliability of "all-green" heatmaps and claims of near-perfect LC performance commonly reported on NIAH benchmarks. Our results demonstrate that even state-of-the-art LLMs possess critical weaknesses in their LC comprehension, and existing evaluation methods are insufficient to reveal them.

Throughout the paper, we detail the construction process of NeedleChain and the associated benchmark schema. Furthermore, we propose a ROPE contraction strategy as a straightforward yet effective approach to enhance LC understanding capabilities. This strategy involves setting the ROPE embedding rotation angle larger during inference than during training, thereby clarifying position distinctions and improving contextual understanding. Through these discussions, we emphasize the need for further exploration to deepen the LC abilities of LLMs.

## 2 NeedleChain

Needlechain is designed as an information-dense context comprehension task. In each context, every sentence contains essential content, requiring the model to fully understand all details for successful completion. Below, we discuss the details related to data construction.

### 2.1 Needle Design

We design our benchmark to analyze the weaknesses of LLMs in processing context by treating the entire context as a single semantic unit and defining a reasoning order required to understand the information provided in the context. To achieve this, we introduce the concept of a "needle" to create connections between preceding and succeeding information. We define two types of needles, considering the smallest unit of information as a sentence, and each needle as a single sentence containing relevant information.

- **Independent Needle** This refers to a sentence that contains independent information without relying on other context. (*"{A} received $1600 last week"*)
- **Dependent Needle** This refers to a sentences that provides information in conjunction with other sentences. It is designed to create an extended context while maintaining a coherent semantic unit. (*"{A} earns twice as much as {B}"* / *"{A} earns half as much as {B}"* / *"{A} earns the same salary as {B}"*)

In particular, we design three types of dependent chains—"halving," "doubling," and "retaining"—that require clear yet straightforward reasoning. This design allows us to evaluate the reasoning capabilities of LLMs without underestimating their core language comprehension abilities due to their lack of precision in complex mathematical calculations, such as decimal operations. We combine these needles according to the specific objectives of our benchmark to form the final evaluation suites.

### 2.2 Chain Composition

Each data point (chain) of **NeedleChain** consists of one independent needle and $k - 1$ dependent needles. By leveraging these needles, the combined chain is designed to maintain a single sequential reasoning order (e.g. *A is related to B, B is related to C, C is related to D*). We then design a question about the last needle in the reasoning order to ensure all context is relevant to the query (*How much salary did {D} get?*). Specifically, we propose three variants as follows. Figure 2 illustrates examples for each chain composition.

- **Forward Chain** The reasoning process must proceed in a **left-to-right** sequence. Accurate conclusions can only be drawn by following the presented order of the given input.
- **Backward Chain** The reasoning process must follow a **right-to-left** sequence. LLM must track the contextual information in the reversely-presented order, starting from the most recently presented data.
- **Mixed Chain** The sequence of required reasoning steps is set **arbitrarily**. The LLM must identify the randomly given reasoning order to arrive at the correct answer.

We design three chain variants using identical needle composition, distinguished solely by the sequence of provided information, thereby ensuring the same reasoning order while different presented order across chain variants. We determine our benchmark's evaluation suite by posing questions related to the needle at the end of reasoning order. This approach ensures our benchmark is structured

so that answering the questions correctly requires 1) following the designated path and 2) fully understanding the given context.

Furthermore, we employ our needles to generate data within the NIAH paradigm, which compiles independent needles into a single context. When we pose a question about one of these independent needles, all other needles not directly related to the question are considered irrelevant context (i.e. haystack.) We denote this benchmark as "NeedleStack (NS)." Through comparing these methods, we effectively demonstrate the LC capabilities that can be assessed using the NeedleChain benchmark.

## 2.3 BENCHMARK DETAILS

In constructing the needle, we utilized a randomly selected name list officially released by the U.S. government [1]. For each data point, we first established a name list, then used this list to create three chain variants and a NeedleStack. This approach allowed us to apply the same name list across our benchmark variants, thereby minimizing unintended bias related to naming Eloundou et al. (2025).

As our benchmark uses synthetic data, we can adjust the context length by increasing the number of needles. Based on this scheme, we investigate the extent of context comprehension in LLMs by examining performance changes as the total number of needles, denoted as $k$, increases to 200. While theoretically we can increase $k$ indefinitely, we find that even with relatively short contexts (token length $\sim$ 2K), we can clearly observe the trends in the language model's LC understanding capacity that we aim to analyze. We generated 200 test instances for each dataset.

## 3 EXPERIMENTS

We conduct experiments using state-of-the-art LLMs reported to possess long context understanding capabilities. In particular, we focus our experiments on the widely used LLMs: Qwen2.5-32B Yang et al. (2024), QwenLong-L1 Yang et al. (2024), Qwen3-32B Yang et al. (2025a), Llama3.3-70B Grattafiori et al. (2024), and GPT-4o Hurst et al. (2024). Detailed information about the models, prompts used for evaluation, and the evaluation environment is provided in the Appendix A.

## 3.1 MAIN RESULTS

We first evaluate the performance of current LLMs on our NeedleChain and NeedleStack benchmarks. The experimental results are presented in Table 1. Key insights from this study are as follows:

**Limitations in Long-Context Understanding** All LLMs show near-perfect performance on the NeedleStack (NIAH paradigm). However, the NeedleChain we designed starts to decline in performance when $k$ exceeds 10, and fails to maintain efficiency when $k$ reaches 50. This indicates a failure to fully comprehend context when its token length is 0.5K. Given that the reported "process-able" context lengths of the LLMs used in our experiments are significantly larger (GPT-4o: 16K, Qwen2.5-32B: 32K, Qwen3-32B: 32K, QwenLong-L1: 1M, Llama3.3-70B: 128K), this provides a clear indication of the limitations in LLMs' LC understanding abilities not revealed by existing benchmarks.

**LLMs Struggle to find solution on Backward Chain** This performance decline was particularly noticeable in the backward chain. It highlights how the LLM's ability to understand context is significantly affected by the reasoning direction, showing vulnerability in reverse reasoning. The backward chain exhibited a greater performance drop compared to mixed chains with arbitrary reasoning paths. The higher performance in mixed chains suggests that what appears to be complex text is no longer a significant issue for LLMs. It indicates that the challenge lies not in the seemingly difficult reasoning paths, but in the requirement for reverse direction reasoning itself.

**Forward-direction reasoning is suitable for LLMs** Among the three chain variants in the NeedleChain benchmark, comprehension in the forward direction is notably high. This suggests

---

[1] https://www.ssa.gov/oact/babynames/decades/century.html

Table 1: Performance of several LLMs on NeedleChain (**NS**: NeedleStack, **F**: Forward chain, **B**: Backward, **M**: Mixed Chain).

| Model | k=5 (Token Length: 0.05K) | | | | k=10 (Token Length: 0.1K) | | | | k=20 (Token Length: 0.2K) | | | |
|---|---|---|---|---|---|---|---|---|---|---|---|---|
| | NS | NeedleChain | | | NS | NeedleChain | | | NS | NeedleChain | | |
| | | F | B | M | | F | B | M | | F | B | M |
| **Qwen3-32B** | 100.0 | 100.0 | 99.5 | 100.0 | 100.0 | 99.5 | 91.5 | 98.0 | 99.5 | 94.0 | 65.0 | 86.0 |
| **Qwen2.5-32B** | 100.0 | 100.0 | 99.5 | 99.5 | 100.0 | 98.0 | 91.5 | 96.5 | 100.0 | 95.0 | 60.0 | 89.5 |
| **QwenLong-L1** | 100.0 | 100.0 | 100.0 | 100.0 | 100.0 | 100.0 | 100.0 | 100.0 | 100.0 | 98.5 | 92.5 | 99.5 |
| **Llama3.3-70B** | 100.0 | 100.0 | 99.5 | 99.5 | 100.0 | 100.0 | 90.5 | 98.5 | 99.5 | 94.0 | 63.5 | 91.0 |
| **GPT-4o** | 100.0 | 100.0 | 100.0 | 99.5 | 100.0 | 100.0 | 95.5 | 98.5 | 100.0 | 98.0 | 78.5 | 88.5 |

| Model | k=50 (Token Length: 0.5K) | | | | k=100 (Token Length: 1K) | | | | k=200 (Token Length: 2K) | | | |
|---|---|---|---|---|---|---|---|---|---|---|---|---|
| | NS | NeedleChain | | | NS | NeedleChain | | | NS | NeedleChain | | |
| | | F | B | M | | F | B | M | | F | B | M |
| **Qwen3-32B** | 99.5 | 80.5 | 13.5 | 46.5 | 99.5 | 68.5 | 10.5 | 10.5 | 99.5 | 47.5 | 1.5 | 4.5 |
| **Qwen2.5-32B** | 100.0 | 81.0 | 19.5 | 44.5 | 100.0 | 65.5 | 7.0 | 16.5 | 100.0 | 43.0 | 0.5 | 8.5 |
| **QwenLong-L1** | 100.0 | 88.5 | 46.0 | 76.0 | 99.5 | 86.5 | 21.5 | 39.0 | 99.5 | 67.8 | 6.5 | 11.0 |
| **Llama3.3-70B** | 100.0 | 76.5 | 31.0 | 79.0 | 100.0 | 67.5 | 22.5 | 45.0 | 99.5 | 44.0 | 18.0 | 12.5 |
| **GPT-4o** | 100.0 | 76.5 | 20.5 | 61.0 | 100.0 | 36.0 | 7.0 | 26.0 | 98.0 | 14.0 | 4.0 | 5.0 |

that when information aligns with the LLM's left-to-right context processing, the model performs optimally. This finding highlights not just the limitations of LLMs but also indicates that presenting information sequentially maximizes their reasoning capabilities.

## 3.2 ERROR ANALYSIS

To gain a deeper understanding of the weaknesses in LLMs' LC comprehension as identified by the NeedleChain benchmark, we analyze the error cases revealed by our benchmark. We categorize the errors in LLMs into three distinct types and find that this taxonomy successfully encompasses all observed error cases:

- **Instruction not Followed**: This refers to instances where the model fails to generate a response by not adhering to the given output format, or fails to determine the final answer.
- **Needle Omission** This refers to cases when certain "needles" are omitted in generating final answer. Specifically, it refers to cases where the name provided as input is absent in the output, resulting in an incorrect answer.
- **Calculation Error** This pertains to situations where the intermediate steps are correct, but an error occurs in the final answer computation. If an error does not fall into the first two categories, we classify such case here.

Concrete examples for each error type are contained in the Appendix. Using this taxonomy, we analyze error cases made by LLMs in our benchmark. Figure 3 displays the results. Key takeaways from our analysis are summarized as follows:

**For small k:** In our study, we identify that the main error factor in the benchmark arises from calculation errors, especially when the number of k is small. This becomes evident when comparing the performance at k=50 using Llama3.3-70B. At k=50, the performance gap between Forward and Backward is approximately 50%, and this discrepancy stems mostly from limitations in computational ability. The diverse tendencies observed even with the same number of k suggest that reasoning direction can impact the computational capacity of LLMs, significantly affecting their understanding performance in LC.

**For larger k:** As we increase k, needle omission emerges as a primary source of error, particularly evident in the Mixed chain. The experimental results demonstrate that as k increases, the decline in performance within this benchmark is mainly attributed to needle omission, which can be regarded as a context missing. We observed that increasing the value of $k$ leads to a loss of "answer generation" objective. This suggests limitations in the LLM's actual LC understanding capability, indicating that achieving a complete understanding remains an ongoing challenge. We provide a more detailed analysis of this phenomenon in a subsequent section.

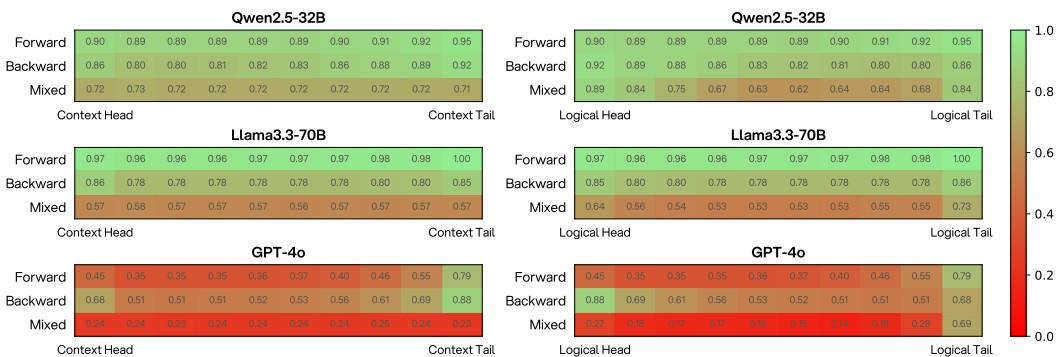

Figure 3: Error analysis on NeedleChain. We analyze errors in each category to determine which of the three predefined error types they fall into.

## 3.3 POSITION HEATMAP

In this section, we utilize NeedleChain to identify positional weaknesses of LLMs in the context comprehension. Note that our input consists of multiple needles, each carrying key information about a specific "name." Considering the final answer can only be derived when the information from all needles is reflected, if any name is omitted from the LLM's response, it indicates that the corresponding needle was not considered during generation.

Through this approach, we identify positional weaknesses by determining whether the names given in the input are included in the generated text. We specifically analyze these positional weaknesses from two perspectives: presented order and reasoning order. We represent the ratio of names mentioned at each position relative to the total number of responses. The experimental results are shown in Figure 4.

Figure 4: Heatmap to show the weaknesses for each position. Left-sided figures shows positional needle-missing heatmap with respect to the "**presented order**". Right-sided figures shows those of "**reasoning order**". We conducted experiments with k=200.

The results indicate that Large Language Models (LLMs) face challenges in fully encapsulating the information provided by the given input, as evidenced by the inconsistent appearance of red spots in the heatmap. This benchmark highlights the **"logically lost-in-the-middle"** phenomenon Liu et al. (2024b). This is clearly observed through the performance in mixed chain. We can find minimal positional weaknesses based on the presented order, as demonstrated by consistent performance declines across all positions. However, when evaluating performance based on reasoning order, it is evident that the model's ability to reflect information significantly diminishes at the "middle position." This finding suggests a practical consideration: LLM lost in the middle of logical flow, rather than in the middle of the given context.

## 3.4 Case Study: Question Variants

The complexity of reflecting information from a given context can vary significantly depending on the question. For instance, when using the context we designed for NeedleStack, a question like "What is the total salary of all mentioned individuals?" requires the language model to integrate all context information. We propose two types of questions to identify performance differences between them. Experimental results are presented in Figure 5.

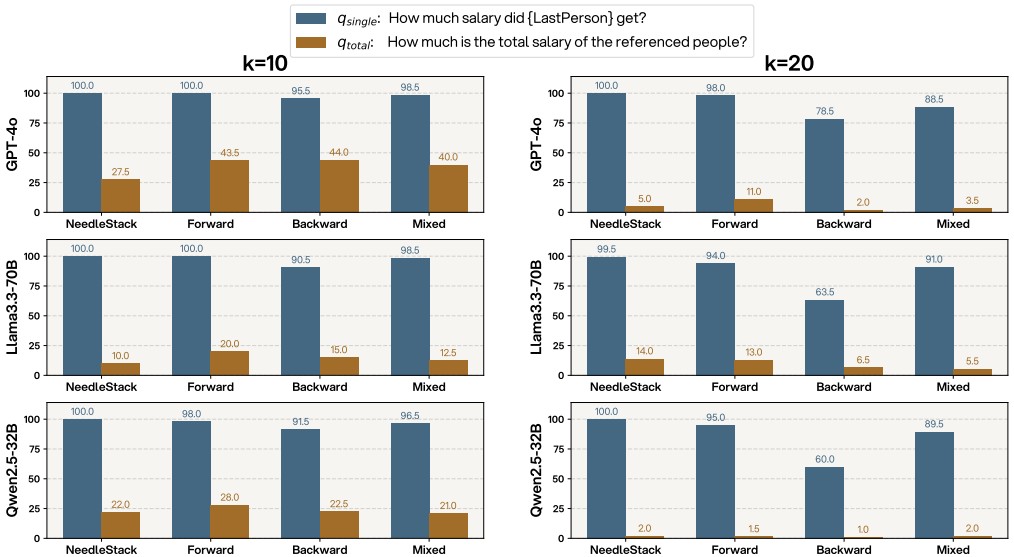

Figure 5: We compare the accuracy of models for different types of questions: those requires understanding the tail of the reasoning chain ($q_{single}$) and those requiring comprehensiv understanding of the entire context ($q_{total}$).

The experimental results demonstrate that when using $q_{total}$ for questioning, the performance of NS significantly declines, with scores dropping below 10 for all LLMs when $k = 20$. This underscores the validity of our constructed data and clearly highlights the limitations in the LLMs' LC capabilities when fully integrating 20 pieces of information.

Notably, when using $q_{total}$, the performance on the NeedleChain benchmark exceeds that of the NS benchmark. This indicates that **deeper semantic connections (*i.e.* reasoning path among context) enhance comprehension more than less interrelated contexts**. These experimental results suggest a practical strategy for enhancing LC comprehension ability of LLMs: Structuring context to emphasize interrelations between information can improve context understanding capabilities. We analyze in the subsequent sections how a reasoning path in the context supports an intact understanding.

## 3.5 Tool Incorporation

In our previous analysis, we identified calculation errors as one of the critical factor contributing to the performance degradation of our benchmark. To address this, recent approaches have sought to mitigate the mathematical limitations of LLMs by incorporating code implementation capabilities Liao et al. (2024); Sharma et al. (2025). Our goal is to assess whether these limitations in language comprehension are underestimated due to computational deficiencies by evaluating performance following code integration. Experimental results are detailed in Table 2.

Interestingly, tool incorporation proved to be particularly effective in NeedleStack, where contextual information between segments is weakly correlated. The ability to improve from a lack of understanding to achieving a high level of performance above 70 points indicates that tool incorporation can be partially beneficial.

However, such incorporation did not prove effective in our NeedleChain benchmark. We observed performance declines in most cases, and in some instances, the decline was even more pronounced.

Table 2: Performance variation on tool merging. We report performance of GPT-4o model.

| Type | Question | k | Code Merging |
|------|----------|---|--------------|
| **NeedleStack** | | | |
| **NIAH** | $q_{total}$ | 5 | **96.0** → 94.0 ( $\nabla$ -2.0 ) |
| **NIAH** | $q_{total}$ | 10 | **27.5** → 53.0 ( $\Delta$ +25.5 ) |
| **NIAH** | $q_{total}$ | 20 | **5.0** → 73.0 ( $\Delta$ +68.0 ) |
| **NIAH** | $q_{total}$ | 50 | **6.5** → 72.0 ( $\Delta$ +65.5 ) |
| **NeedleChain** | | | |
| **Forward** | $q_{single}$ | 50 | **76.5** → 74.0 ( $\nabla$ -2.5 ) |
| **Forward** | $q_{single}$ | 100 | 36.0 → **42.5** ( $\Delta$ +6.5 ) |
| **Backward** | $q_{single}$ | 50 | 20.5 → **23.5** ( $\Delta$ +3.0 ) |
| **Backward** | $q_{single}$ | 100 | **7.0** → 5.0 ( $\nabla$ -2.0 ) |
| **Mixed** | $q_{single}$ | 50 | **61.0** → 54.0 ( $\nabla$ -7.0 ) |
| **Mixed** | $q_{single}$ | 100 | **26.0** → 21.5 ( $\nabla$ -4.5 ) |

This underscores the robustness of our benchmark, demonstrating that it presents challenges that can only be resolved with comprehensive long-context reasoning abilities. This finding suggests that the low performance of LLMs in NeedleChain is not solely due to computational limitations but rather reflects a deficiency in effectively integrating contextual information.

## 4 DISCUSSION

Based on these discussions, we conclude that LLMs do not yet fully comprehend long contexts. There remains considerable room for improvement in processing and understanding given contexts. **We argue that rather than hastily increasing the extent of context length, it might be beneficial to enhance comprehension within a limited range.**

We raise one potential reason for the limited comprehension ability in long contexts as the issue of position separation. As the context lengthens, the ability to distinguish between different parts of the context may deteriorate. To address this, we propose a solution that am-

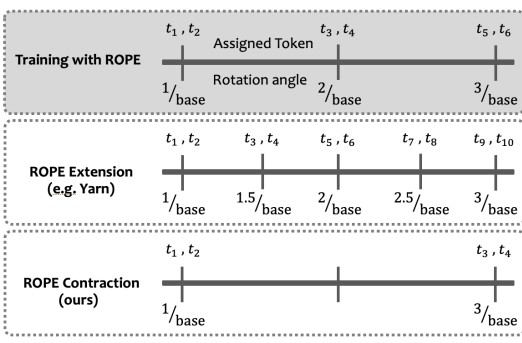

Figure 6: A simple diagram illustrating the concepts of ROPE extension and contraction

plifies positional separation during inference, which we term "ROPE Contraction." A simple example is presented in Figure 6. This approach involves reducing the ROPE base ($\theta$) at inference, which can easily be applied to all LLMs. For our baseline experiment, we tested using half or a quarter of the training $\theta$ during inference. We also compared our results to the commonly used ROPE Extension method known as Yarn Peng et al. (2024). The results of these experiments are shown in Figure 7.

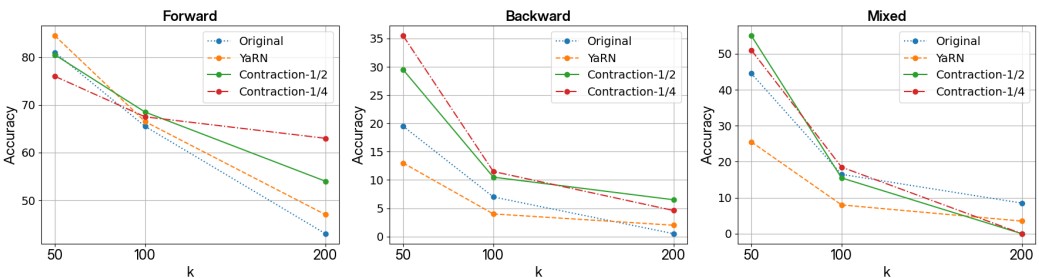

Figure 7: Performance variation derived by the ROPE contraction and extension methodologies

The experimental results clearly support our argument. The use of Yarn significantly decreases performance on the NeedleChain benchmark, while the contraction method substantially improves the ability to understand context intactly. This paper demonstrates that even simple methodologies can lead to meaningful performance improvements, paving the way for future research in this area.

## 5 RELATED WORKS

There are currently numerous approaches to objectively assess the long-context understanding capabilities of LLMs Bai et al. (2024); Hsieh et al. (2024); Kuratov et al. (2024); An et al. (2024a); Li et al. (2024a); Zhang et al. (2024). Among these, the Needle-in-a-Haystack benchmark is a prevailing tool used to evaluate the long-range context understanding of LLMs Yu et al. (2025); Song et al. (2025). However, this benchmark often involves contexts where most information is unnecessary, limiting its ability to assess comprehensive understanding. Consequently, it tends to evaluate only partial and shallow comprehension rather than full understanding of long contexts Hsieh et al. (2024); Kuratov et al. (2024). While attempts like those in Li et al. (2024b) tried to evaluate intact understanding, they often remain rudimentary and fail to thoroughly analyze the potential takeaways from such methods. In particular, these approaches often involve simply identifying ancestors, requiring only shallow reasoning capabilities, which restricts their ability to provide a complete evaluation. In response, we proposed a novel benchmark designed to evaluate the intact understanding of long contexts. Our benchmark and experimental results emphasize that possessing comprehensive understanding within a given context length is more crucial than merely extending the length of the context.

## 6 CONCLUSION

In this paper, we highlight the overestimation of the context understanding abilities of LLMs within the NIAH paradigm. To address this, we introduce NeedleChain, a novel benchmark designed to evaluate the intact context comprehension of LLMs. We design three chain variants to analyze performance variations based on the reasoning order embedded within the context, thereby elucidating the limitations of LLMs' LC understanding capabilities. Our experimental results reveal that even LLMs capable of processing up to 1 million token inputs struggle to fully comprehend information within a 0.5K input length. We also figured out that this difficulty is exacerbated when the reasoning order is set in a backward direction. We further proposed ROPE contraction, a simple yet compelling method to enhance context understanding abilities, achieving significant performance improvements on the NeedleChain benchmark. Our research indicates substantial room for improvement in the LC comprehension capability of LLMs. Additionally, our analysis offers practical advice, suggesting that designing reasoning orders in a forward direction is beneficial when establishing long contexts. For future research, we aim to design benchmarks that encompass a wider range of domains.

## 7 LIMITATION

Due to resource constraints, we were unable to conduct extensive experiments with reasoning models. The excessive length of the input for the reasoning model made it impossible to perform benchmark evaluations under our limited resources. For instance, in a scenario with k=100, the QwQ model required over 30 minutes to process a single query in our experimental setup (our vllm-based environment even flagged such instances as errors). Given the need to process 200 queries for a single test, conducting a wide range of experiments with the inference model was impractical for us. Instead, we report the experimental results for $k$=50 in the appendix and publicly release the data generation code to enable experiments with any higher $k$. We hope this will facilitate broader experimentation using our data in the future.

One limitation of our study is the exclusive use of needles requiring numerical calculation. However, through rigorously designed controlled experiments, we have clearly and robustly demonstrated our conclusions within the given setting. We plan to extend our benchmark paradigm to enhance its generalizability in future research.

## ETHICS STATEMENT

All names referenced in the dataset are fictional, as noted in Section 2.3, and are merely borrowed for illustrative purposes. We affirm that all salaries and names mentioned in the data bear no connection to real-world individuals. Additionally, any potentially referenced names contain no harmful content whatsoever. The models and methodologies employed adhere to community-accepted ethical practices and are not designed for harmful, malicious, or discriminatory purposes. We believe this work aligns with responsible AI research principles and does not introduce foreseeable risks of misuse.

## REPRODUCIBILITY STATEMENT

All datasets and source code are available as detailed in Appendix D. This repository includes the following components:

- The original datasets used in this study
- Source code for constructing the NeedleChain dataset according to any manual parameters (e.g. $k$).
- Evaluation code for replicating all experiments conducted in this study

We publicly release all assets utilized and proposed in our experiments.

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

## A    EVALUATION DETAILS

We conducted all experiments using eight RTX A6000 GPUs. We performed decoding with a temperature setting of 0.6 and a top-p of 0.95, as these parameters represent the optimal prompt suggested in Qwen3[2]. We utilized publicly available checkpoints from HuggingFace Wolf et al. (2020) for all models and executed inference using the vllm Kwon et al. (2023) framework. Configuration for each model is as follows: Qwen2.5-32B Yang et al. (2024) (`Qwen/Qwen2.5-32B-Instruct`), Qwen3-32B Yang et al. (2025a) (`Qwen/Qwen3-32B`), QwenLong-L1 Yang et al. (2025b) (`Tongyi-Zhiwen/QwenLong-L1-32B`), Llama3.3-70B Grattafiori et al. (2024) (`meta-llama/Llama-3.3-70B-Instruct`), QwQ-32B Team (2025) (`Qwen/QwQ-32B`), GPT-4o Hurst et al. (2024) (`gpt-4o-2024-08-06`). The prompt employed for model evaluation is as follows:

Table 3: The default prompt for evaluating NeedleChain

| |
|---|
| **## System Prompt** |
| You are a financial assistant AI skilled in calculating wages and solving salary-related queries. |
| I will give you context with the facts about salary of several people. |
| You need to answer the question based only on the information from the facts. |
| Before you derive the final answer, provide me a brief explanation. |
| Output your final verdict by strictly following this format: '## Answer: ${your_answer}' |
| **## Input Template** |
| There are {num_names} workers in the office. |
| Their names are as follows: {names} |
| |
| Salary for each worker is as follows: |
| {chain} |
| |
| Now, respond to my question: |
| {question} |

## B    PERFORMANCE OF REASONING MODEL

Reasoning models demonstrate exceptional problem-solving for a variety of tasks. In this study, we analyze the performance of reasoning models on our benchmark. The experimental results are presented in Table 4. Along with accuracy, we also report the length of the generated responses. Similar to existing models, reasoning models exhibit diminished performance in backward chain. Although the decline is less pronounced than that in traditional LLMs, we observe a clear trend of performance degradation with larger $k$. This indicates that achieving intact understanding remains a consistent challenge even for reasoning models.

Table 4: Performance of reasoning LLMs on NeedleChain (**NIAH**: Needle Stack, **F**: Forward chain, **B**: Backward, **M**: Mixed Chain). We report both accuracy and the token length of the generated text.

| Model | k=10 (Token Length: 0.1K) | | | | k=20 (Token Length: 0.2K) | | | | k=50 (Token Length: 0.5K) | | | |
|---|---|---|---|---|---|---|---|---|---|---|---|---|
| | NS | NeedleChain | | | NS | NeedleChain | | | NS | NeedleChain | | |
| | | F | B | M | | F | B | M | | F | B | M |
| Qwen2.5-32B | 100 (36.41) | 100 (283.76) | 91.5 (284.09) | 93.5 (306.14) | 100 (33.27) | 97 (500.44) | 53 (505.64) | 76 (568.13) | 99.5 (31.98) | 84.5 (1072.82) | 13 (1175.93) | 25.5 (1282.09) |
| QwQ-32B | 100 (300.95) | 100 (618.09) | 99.5 (1220.63) | 99.5 (934.02) | 99.5 (626.59) | 93.5 (1036.64) | 76 (2950.03) | 91.5 (2196.24) | 100 (875.13) | 86.5 (2727.29) | 19 (7938.16) | 62.5 (6822.31) |
| Qwen3-32B | 100 (283.97) | 100 (492.95) | 99 (573.08) | 100 (655.75) | 100 (339.8) | 97 (826.15) | 88.5 (1031.43) | 96.5 (1141.44) | 98.5 (422.35) | 77.5 (1810.63) | 24 (3001.62) | 63 (3368.24) |

---

[2]https://huggingface.co/Qwen/Qwen3-32B

## C    LLM Usage Disclaim

We utilized an AI writer solely for polishing the English expressions in the paper. Beyond this, no assistance was received in terms of ideation or other content development; the AI writer's role was strictly limited to correcting English expressions.

## D    Temporal Repository

Source code and datasets are available at `https://anonymous.4open.science/r/NeedleChain-5954`.

