# OpenReview forum: "NeedleChain: Measuring Intact Long-Context Reasoning Capability of Large Language Models"
_ICLR.cc/2026/Conference — ICLR 2026 Conference Withdrawn Submission_

### Official Review · Reviewer_wQTb · 2025-10-29

**Soundness:** 2
**Presentation:** 2
**Contribution:** 2
**Rating:** 2
**Confidence:** 4

**Summary:**

This work highlights that existing Needle-in-a-Haystack benchmarks fail to faithfully evaluate the long-context understanding ability of large language models (LLMs). To address this, the authors propose the NeedleChain dataset, which manually constructs contexts involving complex multi-hop reasoning. Specifically, it introduces two data modes—Independent Needle and Dependent Needle—and three logical formats—Forward Chain, Backward Chain, and Mixed Chain—to test LLMs’ comprehension of three mathematical relations: halving, doubling, and retraining. Experimental results show that advanced models such as Qwen2.5-32B, Llama3.3-70B, and GPT-4o exhibit performance degradation on NeedleChain as the token length increases, suggesting that longer context windows do not necessarily imply better contextual reasoning. The authors therefore call for future research to focus more on genuine long-context understanding rather than merely expanding context length. In addition, they design a novel ROPE strategy to further enhance LLMs’ reasoning capability.

**Strengths:**

1. The topic of this study is important. The limitations of the Needle-in-a-Haystack evaluation scheme have gradually become common sense in the community. Exploring improved methodologies for assessing LLMs’ long-context understanding is therefore highly valuable.

2. The construction of the dataset is clear and flexible. The results are reproducible, and the context length can be easily extended.

3. The experimental findings convincingly demonstrate that current LLMs still lack genuine long-context comprehension ability.

**Weaknesses:**

1. The evaluation setting deviates from the stated goal. Although the paper emphasizes long-context reasoning, the proposed NeedleChain dataset focuses too heavily on reasoning itself, while the maximum context length is only around 2k tokens, which is far below the 32k+ scale typically considered in long-context evaluation. Moreover, the task setting is overly narrow, involving only simple proportional arithmetic relations (0.5×, 1×, 2×) about character salaries. In contrast, benchmarks like LongBench adopt more diverse scenarios to comprehensively evaluate long-context understanding.

2. The proposed ROPE Contraction method is unclear. The description lacks sufficient detail. Since modifying RoPE is a common strategy for extending context windows, it is not clear how the proposed approach differs from prior work or why it theoretically improves reasoning, especially when the evaluated context length is relatively short.

3. Numerous typos and formatting issues. For example, the content of Fig. 2 does not match its caption; the abbreviation should be ROPE rather than RoPE; and citation commands should follow LaTeX conventions, using \citep{} for parenthetical citation and \citet{} when used as the grammatical subject.

**Questions:**

Please see the weakness part

---

### Official Review · Reviewer_jZFm · 2025-10-30

**Soundness:** 1
**Presentation:** 1
**Contribution:** 1
**Rating:** 2
**Confidence:** 5

**Summary:**

This paper introduces a benchmark for evaluating the long-context performance of LLMs using fact chaining, or as they call it, NeedleChain. Unlike the well-known Needle-in-a-Haystack benchmark, NeedleChain provides a sequence of interrelated statements, where answering the final query requires reasoning over all the given facts. Experimental results across various models show that as the number of statements within the context window increases, the task becomes increasingly challenging. The paper also introduces a ROPE contraction mechanism, claiming it contributes to performance gains.

**Strengths:**

While the experiments are not conducted in a fully controlled setup, the paper does include a fair amount of analysis—for example, examining error types and positional effects.

**Weaknesses:**

Novelty:
Although the paper references some relevant works, it does not adequately compare its contributions with prior research. For instance, RULER includes a variable tracking (VT) task where multiple statements are chained together. Due to its controlled setup—where the number of statements is fixed regardless of haystack length—it provides a more consistent measure of task complexity (see next point). Moreover, Li et al. (2024b) introduce multi-needle reasoning, which is highly similar to NeedleChain.


Task complexity varies with length:
Increasing the number of statements simultaneously increases both the context length and the task’s inherent complexity. This makes it difficult to determine whether model failures are due to limitations in length generalization or in problem complexity. In RULER, for example, a VT task could be defined with eight interconnected facts while filling the remaining context with haystack text—allowing evaluation of long-context performance at a fixed difficulty level.


Using arithmetic:
Arithmetic is already a challenging task for LLMs, independent of context length (Yuan et al., 2023). Moreover, in cases with k=200, results can be as high as 2**200, introducing severe numerical precision and overflow issues. Even with tool use, such decimal computation errors would remain.


Unclear and insufficient evaluation of the ROPE contraction method:
Figure 6 is unclear, and the explanation fails to properly describe how ROPE contraction works. Does it apply across all frequencies? In contrast, YARN uses an NTK-by-parts approach to preserve high-frequency (short-distance) relationships, as noted by Peng et al. Therefore, the observed gap at k=50 between YARN and the original model in Figure 7 is questionable, since the context at that length is relatively short (~2K tokens).
Yuan et al., 2023: How Well Do Large Language Models Perform in Arithmetic Tasks?


Other Issues:
- The original NIAH task is cited incorrectly (lines 052, 440, 443):
Kamradt, G. (2023). Needle in a Haystack – Pressure Testing LLMs. GitHub Repository, p.28.
- Line 253 mentions that examples for each error type are discussed in the appendix, but they are not included.
- In the main experiment, the models exhibit some level of in-context reasoning. Based on the prompt shown in the appendix, the models produce a brief explanation followed by a final answer, essentially resembling chain-of-thought (CoT) prompting.
- The error analysis relies on the limited explanations output by the model, which may not accurately reflect its internal (implicit) reasoning processes.
- Several experimental details remain unclear. For example:
 Which model was used for Figure 7?
 What is the YARN scaling factor?
- The needle templates are questionable. For example:
 				{A} received X last week.
The query then asks about salary (How much salary did {Z} get?), which may refer to a different timeframe and thus be ambiguous for the model.



Missing References:
- Random-Access Infinite Context Length for Transformers (Amirkeivan Mohtashami, Martin Jaggi, 2023).
- Is It Really Long Context if All You Need Is Retrieval? Towards Genuinely Difficult Long-Context NLP – addresses the issue described in Weakness No. 2.
- NOLIMA: Long-Context Evaluation Beyond Literal Matching – discusses the backward reasoning challenge
- Michelangelo: Long-Context Evaluations Beyond Haystacks via Latent Structure Queries.

**Questions:**

(no questions)

---

### Official Review · Reviewer_BuXb · 2025-10-30

**Soundness:** 2
**Presentation:** 2
**Contribution:** 2
**Rating:** 4
**Confidence:** 5

**Summary:**

This paper critiques the popular "Needle-in-a-Haystack" benchmark, arguing it fails to measure true long-context understanding in LLMs, focusing instead on simple information retrieval. The authors introduce NeedleChain, a new benchmark requiring models to integrate and reason over the entire input. Experiments reveal a significant gap between LLMs' ability to process long texts and their capacity for genuine comprehension.

**Strengths:**

1. This work provides a detailed error analysis and case study of the proposed benchmark, uncovering some interesting experimental findings.
2. This work introduces NeedleChain, a novel task that challenges even advanced LLMs, thus offering a new angle for a more complete assessment of their abilities.

**Weaknesses:**

1. Lacking experimental results from more powerful closed-source models such as Claude Sonnet 4.5, Gemini 2.5 Pro, and GPT-5 to prove the necessity of NeedleChain.
2. In Table 1, the longest context length in the main experiment is only 2k. This is actually evaluating the model's logical reasoning ability rather than its long-context modeling capability. The authors could include additional experimental results using reasoning models to examine whether stronger reasoning capabilities can improve accuracy.
3. The NeedleChain task deviates from real-world long context application scenarios. Even achieving strong performance on this task does not demonstrate true long context modeling abilities.
4. ROPE Contraction is not actually a new method, as similar attempts have already been made in prior work [3].
5. Missing citations for some long-context modeling benchmark works [1][2].


[1] BAMBOO: A Comprehensive Benchmark for Evaluating Long Text Modeling Capacities of Large Language Models. (COLING'24)

[2] Leave No Document Behind: Benchmarking Long-Context LLMs with Extended Multi-Doc QA. (EMNLP'24)

[3] Scaling Laws of RoPE-based Extrapolation. (ICLR'24)

**Questions:**

1. Regarding the Error analysis in Figure 3, it is unclear how to handle cases when both Needle Omission and Calculation Error occur simultaneously.

---

### Official Review · Reviewer_Pupg · 2025-11-01

**Soundness:** 3
**Presentation:** 3
**Contribution:** 2
**Rating:** 4
**Confidence:** 2

**Summary:**

This paper introduces NeeleChain, a benchmark designed to evaluate the intact context comprehension of LLMs. Three chain variants (forward, backward, and mixed) are designed to analyze performance variations based on the reasoning order. Experimental results reveal the gap between LLMs’ ability to process long inputs and their capacity for full understanding the context.

**Strengths:**

+ a new benchmark using chained inputs to evaluate the intact context comprehension of LLMs
+ experimental findings to identify the limitations of LLMs’ long-context understanding capabilities

**Weaknesses:**

- the benchmark focuses on numerical calculation only and thus is limited
- the findings are kind of expected, making the research contributions weak

**Questions:**

Is the poor performance of the LLMs over NeedleChain because their weak numerical calculation ability rather than the intact context comprehension of the LLMs?

---

### Note · Authors · 2025-11-26

I have read and agree with the venue's withdrawal policy on behalf of myself and my co-authors.